# Electromagnetic Imaging of Passive Intermodulation Sources Based on Virtual Array Expansion Synchronous Imaging Compressed Sensing

**Siyuan Liu** [1,2], **Musheng Liang** [1,2], **Zihan Cheng** [1,2], **Xinjie Li** [1,2], **Menglu Ma** [1,2], **Feng Liang** [1,2] and **Deshuang Zhao** [1,2,*]

1   Yangtze Delta Region Institute (Huzhou), University of Electronic Science and Technology of China, Huzhou 313001, China
2   Institute of Applied Physics, University of Electronic Science and Technology of China, Chengdu 611731, China
*   Correspondence: dszhao@uestc.edu.cn

**Abstract:** Interference caused by passive intermodulation (PIM) has seriously affected system performance. Compressed sensing (CS) can be used to locate radial PIMs. In CS-based electromagnetic imaging methods, the ability to locate the number of targets is usually limited by the number of receiving antennas, and there are flooding problems in the imaging results. To solve these problems, traditional methods are used to increase the number of receiving antennas. To reduce the number of receiving antennas and improve the imaging effect, this paper proposes a virtual array expansion synchronous imaging compressed sensing (SI-CS) method. With the full-wave electromagnetic simulation demonstrations, the proposed method exhibits significant improvement in imaging performance compared to the conventional methods. In addition, the proposed method can discriminate between targets in the presence of the flooding problem in which the paraflap is too large and the location of the PIM source cannot be determined by the amplitude value.

**Keywords:** compressed sensing; passive intermodulation; electromagnetic imaging; coprime array; fuzzy synchronous principle

## 1. Introduction

With the upgrading of wireless mobile communications, base station antennas have developed rapidly, from the omnidirectional base station antennas in the 1G and 2G eras to the directional base station antennas in the 3G and 4G eras. In the 5G era, phased array antennas have been widely used in base station antennas due to their multifrequency and multibeam characteristics. However, passive intermodulation (PIM) in phased array antennas is a problem that needs to be solved urgently. Due to the multichannel, multifrequency, high-power transmitting branch inside phased array antennas, when high-power two-frequency signals pass through the phased array antennas and the passive components (such as connectors, cables, etc.), it is easy to produce a PIM signal, eventually leading to the severe degradation of the receiver. In order to find the position of the PIM generated by the phased array, a fast and accurate electromagnetic imaging method is quite desirable.

Compressed sensing (CS), also known as compressive sampling or sparse sampling, is a technique for finding sparse solutions to underdetermined linear systems. In the signal processing field, it is well-known from the Nyquist sampling theorem that if the highest frequency of the signal is less than half of the sampling frequency, the original signal can be perfectly recovered from the sampling result. However, it becomes possible in the CS theory for the signal to be recovered with fewer samples under the circumstance of known sparsity. The CS theory is widely utilized in many applications, such as signal

processing [1–4], image processing [5–9], magnetic resonance imaging (MRI) [10–14], and target imaging [15–25].

Electromagnetic imaging localization based on the CS algorithm has been attractive during recent years. By applying CS to radar imaging, the analog-to-digital converter can be removed at the receiver side, which significantly simplifies the radar imaging system [26]. By applying CS to ultrawideband signals with a sparse random matrix and randomly selected frequency points in a random sampling antenna, it becomes very easy to implement imaging algorithms in hardware [27,28]. Although the signal waveforms acquired at different locations are different in most imaging systems, their contours have certain similarities. By using the existence of certain group structure features in the data, CS-based group sparse reconstruction algorithms are proposed [29–31]. A received signal strength localization protocol is proposed. Under this protocol, we can exploit the spatial correlations among the received measurements to jointly estimate the positions of mobile users [32]. A combination of spectrum sensing and localization tasks is observed for the first time. Additionally, a simulation shows promising and interesting results for compressed sensing techniques applied to this formalism. Cognitive radio systems are able to communicate in the available bands in the available directions [33]. Diffusive compressed sensing (DCS) is proposed to simplify the measuring devices and improve the devices' resolution. Additionally, DCS has better quality compared to the case of classic CS and has comparable quality to the case of dense sampling [34]. Exact reconstruction is feasible with spatially undersampled data. Additionally, we can reduce the spatial sampling density by linearly increasing the temporal one, allowing for the lower cost of the sensor network [35].

In general, the number of receiving antennas affects the performance of the imaging methods. More receiving antennas can increase the number of targets and positioning accuracy. However, due to the limited space and cost considerations, it is not possible to increase the number of receiving antennas indefinitely. To solve this problem, virtual array expansion technology such as coprime array decomposition provides an alternative approach. In [36], a method of direction of arrival (DOA) estimation with multiple signal classification (MUSIC) algorithms using the coprime array is performed, and the results show that the closest peak from the spatial spectrum is resolved. Subsequently, the idea of coprime array decomposition is extended from a one-dimensional array to a two-dimensional one to estimate pitch and azimuth angles [37].

In this paper, we propose a synchronous imaging compressed sensing (SI-CS) method, based on the fuzzy synchronous principle and coprime array, which can effectively image and locate PIM sources in different states with a small number of receiving antennas. The proposed method adopts the array arrangement technique of the coprime array, effectively reduces the hardware cost, and strengthens the imaging localization effect by constructing virtual receiving antennas. PIMs in phased arrays are often generated simultaneously at different locations. In the imaging results, certain peaks appear at different frequency points of the PIM source location, while secondary flaps are generated randomly. After filtering out the parts larger than a certain value, the results of each frequency point are normalized and multiplied together. Due to the fuzzy synchronous principle, the PIM source position will be highlighted and the subflap position will be reduced. Compared with the traditional idea of adding up the calculation results at each frequency point, the proposed method can effectively solve the problem of part of the PIM source being flooded by the secondary flap. Additionally, at the same time, the imaging of different PIM sources in the results is enhanced. The simulation considers the presence of a different number of PIM sources. Additionally, corresponding simulation data are used to analyze the performance of the proposed method. The simulation results verify the practicality of the method. Compared with the existing work, the main contributions are listed in the following.

(1) The proposed method can solve the problem of the large peak area at the PIM source location in the imaging results and can obtain imaging results with smaller peak areas with smaller computational effort.

(2)  The proposed method solves the problem that a smaller number of antennas cannot locate multiple PIM sources. The method solves the flooding problem with higher localization accuracy.

(3)  The proposed method solves the problem of the low energy share of the PIM source location in the imaging results. The solution to this problem can make the PIM source location in the imaging results have more focused and more accurate positioning.

## 2. Signal Model for Detecting PIMs

### 2.1. Coprime Array

The coprime array is a nonlinear array. The array is formed by combining two subarrays. The number of elements of the two subarrays satisfies the mutual quality relationship. One antenna array has $N$ receivers and another antenna array has $M$ receivers. There is a mutual-quality relationship between $M$ and $N$. A one-dimensional coprime array shown in Figure 1 is formed by the combination of two uniform arrays. Subarray arrays with $N$ receivers are spaced Md apart, and subarray arrays with M receivers are spaced Nd apart. The two subarrays share the first array element. Then, the coprime array has a total of $M + N - 1$ array elements in each half coordinate. The position of the array elements can be represented by the set $L$.

$$L = \{Nmd, 0 \le m \le M - 1\} \cup \{Mnd, 1 \le n \le N - 1\} \tag{1}$$

Similarly, repeating this process in the other direction results in a two-dimensional coprime array. Using this coprime array to receive PIM signals from $K$ different locations, the received data for the array element with coordinates $u_i(u_i \in L, i = 1, 2, \ldots, M + N - 1)$ can be expressed as:

$$\boldsymbol{r}_{u_i}(\omega) = \sum_{k=1}^{K} \boldsymbol{s}_k(\omega)\boldsymbol{h}_k(\omega) + \boldsymbol{n}_{u_i}(\omega) \tag{2}$$

The matrix form can be expressed as:

$$\boldsymbol{r}(\omega) = \mathbf{H}(\omega)\boldsymbol{s}(\omega) + \boldsymbol{n}(\omega) \tag{3}$$

where $\boldsymbol{r}(\omega) = [\boldsymbol{r}_1(\omega), \boldsymbol{r}_2(\omega), \ldots, \boldsymbol{r}_{M+N-1}(\omega)]^T$ is the received signal of the antenna array. $\boldsymbol{s}(\omega) = [\boldsymbol{s}_1(\omega), \boldsymbol{s}_2(\omega), \ldots, \boldsymbol{s}_K(\omega)]$ is the K signals. $\boldsymbol{n}(\omega)$ is the noise received by the antenna array, and $\mathbf{H}(\omega) = [\boldsymbol{h}_1(\omega), \boldsymbol{h}_2(\omega), \ldots, \boldsymbol{h}_K(\omega)]$ is the transfer function matrix from the PIM sources' location to each receiving antenna.

From the known coprime array received signal, the autocorrelation matrix can be calculated for it to obtain:

$$\begin{aligned}
\mathbf{R}' &= \mathrm{E}[\boldsymbol{r}(\omega)\boldsymbol{r}^{\mathrm{H}}(\omega)] \\
&= \sum_{k=1}^{K} \sigma_k^2 \boldsymbol{h}_k(\omega)\boldsymbol{h}_k^{\mathrm{H}}(\omega) + \sigma_n^2 \mathbf{I}
\end{aligned} \tag{4}$$

where $\sigma_k^2$ is the transmitting power of the kth signal. Additionally, $\sigma_n^2$ is the noise power.

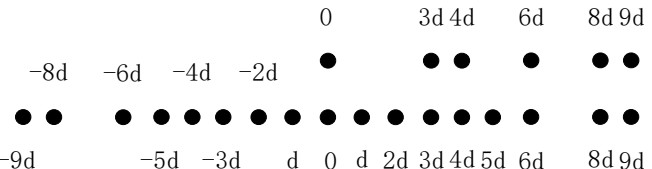

**Figure 1.** The structure of the coprime array with $M = 3$, $N = 4$, and the virtual array constitute.

The signal of the virtual array can be obtained using the received signal of the actual array element of the coprime array. The array formed by the newly obtained received

signals is called a virtual array. The structure of a one-dimensional coprime array with $M = 3$, $N = 4$, and the virtual array constitutes is shown in Figure 1.

The vectorization of the autocorrelation matrix is performed. Then, the virtual received signal can be obtained by:

$$z = vec(\mathbf{R}') = \mathbf{H}' s' + \sigma_n^2 \bar{\mathbf{I}} \tag{5}$$

where $vec(\bullet)$ is the vectorization matrix, which arranges the columns of the matrix into a column vector in order. $s'$ is the source of the virtual received signal. $\bar{\mathbf{I}} = vec(\mathbf{I})$ arranges the columns of the unit matrix into a column vector in order. $\mathbf{H}'$ is the transmission matrix of the virtual received signal. $\mathbf{H}'$ can be expressed as:

$$\mathbf{H}'(\omega) = h_k(\omega) \otimes h_k^*(\omega) \tag{6}$$

where $\otimes$ is Kronecker's product.

In this paper, we use the T-shaped coprime array shown in Figure 2. The equivalent model in Figure 2 is based on the potential PIM shielded in the structure-known devices. Additionally, assume that the signal emitted by the PIM source is noncoherent. The black dot in Figure 1 represents the receiving antenna array and the red dot represents the location of the PIM targets. Additionally, the model is placed in a free space. Hence, there are $2(M + N - 1) - 1$ antennas in the z-direction and $M + N - 1$ antennas in the x-direction.

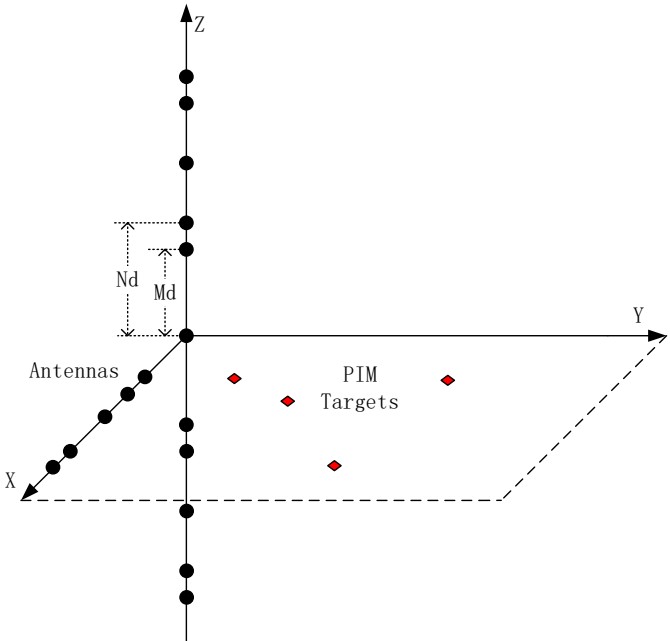

**Figure 2.** The diagram of detecting PIMs based on CS imaging.

*2.2. UWB-CS*

According to the signal model described earlier, the received signal of UWB-CS can be expressed as:

$$\mathbf{R}_{UWB-CS}(\omega) = \mathbf{H}(\omega, l)\mathbf{S}(\omega) + \mathbf{N}(\omega) \tag{7}$$

The receiving antenna array receives K incident signals. We obtain the full transfer function information of the desired imaging surface using numerical calculations or simulations. The transmission matrix from each receiving antenna unit to each point of the imaging surface is constructed from the acquired transmission functions. Sparsification of the received signal can be expressed as:

$$\mathbf{R}_{UWB-CS}(\omega_i) = \mathbf{H}(\omega_i)x(\omega_i) + \mathbf{N}(\omega_i) \tag{8}$$

where $\mathbf{H}(\omega)$ is the transmission matrix from each pixel point on the imaging plane to the receiving antenna. The dimension of $\mathbf{H}(\omega)$ is the number of several receiving antenna array cells multiplied by the number of pixel points on the imaging surface. $x(\omega)$ is the transmitted signal of each pixel point. The dimension of $x(\omega)$ is the number of pixel points multiplied by one. $\mathbf{N}(\omega)$ is the noise received by the antenna array.

Since the number of pixels on the imaging surface is much larger than the number of PIM sources, the sparsification condition is satisfied. We can deal with this problem by minimizing the $l_0$-norm of $x(\omega_i)$, i.e., $\|x(\omega_i)\|_0$, subject to $\|\mathbf{R}_{UWB-CS}(\omega_i) - \mathbf{H}(\omega_i)x(\omega_i)\|_2 \leq \varepsilon$, where $\varepsilon$ is a specific noise allowance. This problem can be expressed as:

$$
\begin{aligned}
&\min\|x(\omega_i)\|_0 \\
&s.t\|\mathbf{R}_{UWB-CS}(\omega_i) - \mathbf{H}(\omega_i)x(\omega_i)\|_2 \leq \varepsilon
\end{aligned}
\tag{9}
$$

We can replace $l_0$-norm by $l_1$-norm to solve this problem. The $l_1$-norm equation can be represented as:

$$
\begin{aligned}
&\min\|x(\omega_i)\|_1 \\
&s.t\|\mathbf{R}_{UWB-CS}(\omega_i) - \mathbf{H}(\omega_i)x(\omega_i)\|_2 \leq \varepsilon
\end{aligned}
\tag{10}
$$

The above equation calculates the imaging results at only one frequency point. Assume that there are n frequency points in the whole bandwidth. To obtain imaging results at a certain bandwidth which is UWB-CS, we can proceed as follows:

$$
\mathbf{X}_{UWB-CS} = \sum_{i=1}^{n} x(\omega_i)
\tag{11}
$$

## 3. The Proposed Improved CS Method

### 3.1. Transmission Matrix Frequency Selection

When the selected imaging surface is large and there are more points on the imaging surface, the computation time of sparse reconstruction is significantly enhanced. It is a reasonable choice to select the appropriate frequency points for the sparse reconstruction calculation. When the frequency interval is small, the results obtained by sparse reconstruction will be similar. These results can be considered redundant results. Therefore, a covariate, $\rho$, is defined to represent the correlation coefficient of the transmission matrix between different frequency points.

$$
\rho_q = \frac{\left|\mathbf{H}'(\omega_c) - \mathbf{H}'(\omega_q)\right|^2}{\max(\left|\mathbf{H}'(\omega_c) - \mathbf{H}'(\omega_q)\right|^2)}
\tag{12}
$$

where $\omega_c$ is the center frequency. The larger correlation factor makes more differences between different transmission matrices. Hence, the threshold value, $\delta$, is studied to remove the redundant statistical data.

$$
\rho_q \geq \delta
\tag{13}
$$

If the correlation coefficient is greater than the set covariance, $\delta$, the matrix, $\mathbf{H}'(\omega_q)$, is obtained as the expected matrix.

### 3.2. Fuzzy Synchronous Principle

The imaging results are not consistent at different frequency points. However, there is a peak at the PIM source location, and the size of the peak varies with the power; that is, the location of the PIM sources at different frequency points will exist in the spatially fuzzy synchronization. Considering the presence of noise, the imaging results are not flat. Define a covariate, $\mu$, related to the signal power.

$$
x_0 \geq \mu
\tag{14}
$$

where $x_0$ represents the points in the imaging results at the center frequency point.

When the calculation result of a point is greater than that of the set parameters, it is considered that there may be a PIM source here. Normalization is performed separately in each screened range. At this point, the power at the peak of the location of all possible PIM sources is unified. Regardless of the magnitude of the change in PIM positions or in larger subflaps brought about by noise effects, they are expressed as the same intensity.

$$x_{0_p}{}' = \left|x_{0_p}\right| \bigg/ \max\left|x_{0_p}\right|, p = 1, 2, \ldots, P \tag{15}$$

$x_{0_p}$ represents the range of the pth possible PIM source.

At this point, the imaging results corresponding to the central frequency point processed by (11) can be expressed as:

$$x_0{}' = \left\{x_{0_1}{}', x_{0_2}{}', \ldots, x_{0_P}{}' \& \text{other}\right\} \tag{16}$$

From (11), we can see that the effect of different power PIM sources on the imaging results is eliminated in the results, effectively eliminating the possibility of some signals being drowned out in UWB-CS.

### 3.3. SI-CS Method

For the filtered frequency points, the compressed sensing algorithm can obtain the results of their corresponding frequency points. The calculation is performed according to the following:

$$\min\|x(\omega_i)\|_1$$
$$\text{s.t} \|z'(\omega_i) - \mathbf{H}'(\omega_i)x(\omega_i)\|_2 \leq \varepsilon \tag{17}$$

where $i$ represents the $i$th frequency point. $\varepsilon$ is a parameter related to noise.

The calculation results of each filtered frequency point are further calculated by using the fuzzy synchronization principle. At this time, the calculation results of each frequency point are obtained as $x'(\omega_0), x'(\omega_1), \ldots, x'(\omega_n)$. The formula for SI-CS can be expressed as follows:

$$x_{\text{SI-CS}} = \prod_{i=1}^{n} x'(\omega_i) \tag{18}$$

### 3.4. Quantitative Indicators for Imaging Quality

In this section, the quantitative indicators are defined to evaluate the imaging performance. By assuming that the localization results of the $K$ target centers are $(x_k, y_k)$, $k = 1, \ldots, K$, the localization error is given by:

$$RMSE = \sqrt{\frac{1}{K}\left(\sum_{1}^{K}(x_k - x_{real-k})^2 + (y_k - y_{real-k})^2\right)} \tag{19}$$

where $(x_{real-k}, y_{real-k}), k = 1, \ldots, K$ is the real position of the $K$ target centers.

The energy share of the PIM source location in the imaging results is also a factor in assessing the imaging quality. The calculation method is as follows:

$$E_s = \frac{E_1 + E_2 + \cdots + E_K}{E} \times 100\% \tag{20}$$

where $E_k$ is the energy of the kth PIM source in the imaging result, $E$ is the total energy of the imaging result. $E$ is related to $x^2$.

## 4. Simulation Results and Discussion

In this section, the full-wave electromagnetic simulation for the assumed PIM source signal is conducted to evaluate the effectiveness of the proposed method. In the full-wave electromagnetic simulation model, the T-shape in Figure 2 is chosen for the coprime array, where $M = 3$, $N = 4$, 16 arrays in total. The simulated PIM signals are generated by CST. Simulations for a single target and multiple targets are conducted in free space of $300 \times 300$ mm$^2$. PIM sources emit the Gaussian pulses of which the center frequency is 1.7 GHz and bandwidth is 200 MHz. The efficiency is examined in terms of CPU time of all the localization algorithms on a PC with AMD R5 5600× and 32 GB RAM in the same environment.

To assess the applicability of the proposed method, the SNR of the received waves and several common factors that may degrade the imaging and localization results are considered separately in the following simulations. In the proposed SI-CS scheme, the transmission matrix frequency selection method is used to reduce computational effort while ensuring the stability of results. Additionally, the fuzzy synchronous principle is used to filter and enhance the imaging results of PIM sources. For comparison, the conventional UWB-CS method is also applied to locate the PIM sources. The imaging results from the conventional UWB-CS method and our proposed SI-CS method are compared. In the following examples, we will consider the imaging of a single, double, and multiple PIM source targets in sequence.

### 4.1. A Single Target

In this example, an imaging area of $300 \times 300$ mm$^2$ size is set. A PIM source is set in the center. The coordinate of the PIM source is (150 mm, 250 mm). The frequency interval is set to 10 MHz. In addition, white Gaussian noise was added to check the performance of the proposed method. The mean value of the added noise is 0, and the variance is the average power of the noise.

Based on Equation (12), $\rho_q$ values at different frequencies are shown in Figure 3. From the result, we can know that the difference between the transmission matrix closer to the center frequency in frequency and the transmission matrix corresponding to the center frequency is smaller, and at this time the calculation of parameter 1 results in a value of only around 0.1–0.3. Hence, we choose as 0.8 to be $\delta$. We can utilize it for 1.6 GHz, 1.61 GHz, 1.7 GHz, 1.79 GHz, 1.8 GHz.

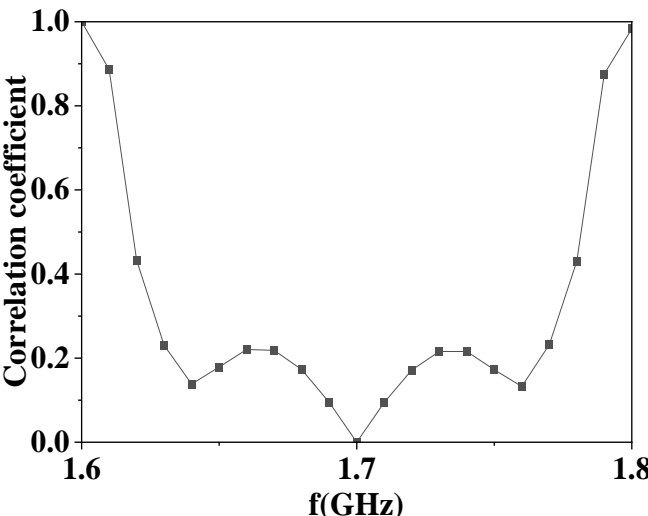

**Figure 3.** Correlation coefficient of the transmission matrix for a single PIM source.

The center frequency obtained through the imaging results is shown in Figure 4.

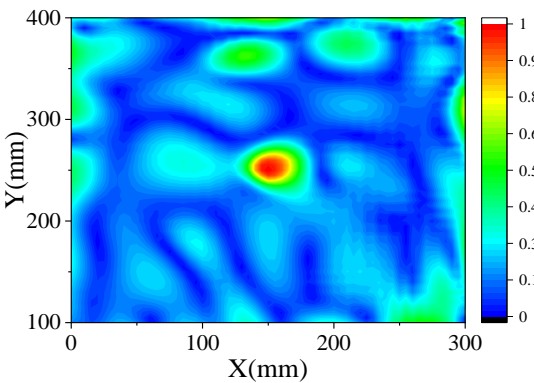

**Figure 4.** Imaging results of center frequency for a single PIM source.

As we can see in Figure 4, there is a region of maximum peak in this imaging result, and there is no paraflap comparable to the maximum peak point. In order to filter out the presence of PIM sources at the remaining locations, we set $\mu$ to 0.5, which is the half-peak of the normalized maximum.

The imaging results of UWB-CS and SI-CS obtained from the sensor data with SNR = 20 dB are shown in Figure 5. From the results (a) and (b), we can see that both methods can accurately locate the set PIM source. However, the results obtained by the SI-CS method have a smaller imaging spot range at the PIM source location. At the same time, the SI-CS method obtains imaging results with smaller paraflaps and smoother imaging results. The computation time of UWB-CS and SI-CS is 717.49 s and 173.08 s. The trend of the peak in the lateral direction at the PIM source location for the UWB-CS method and the SI-CS method is plotted in Figure 6. By comparison, it can be found that the area of the SI-CS peak imaging section is only 16% of that in the UWB-CS method. This allows SI-CS to characterize the PIM source location more precisely.

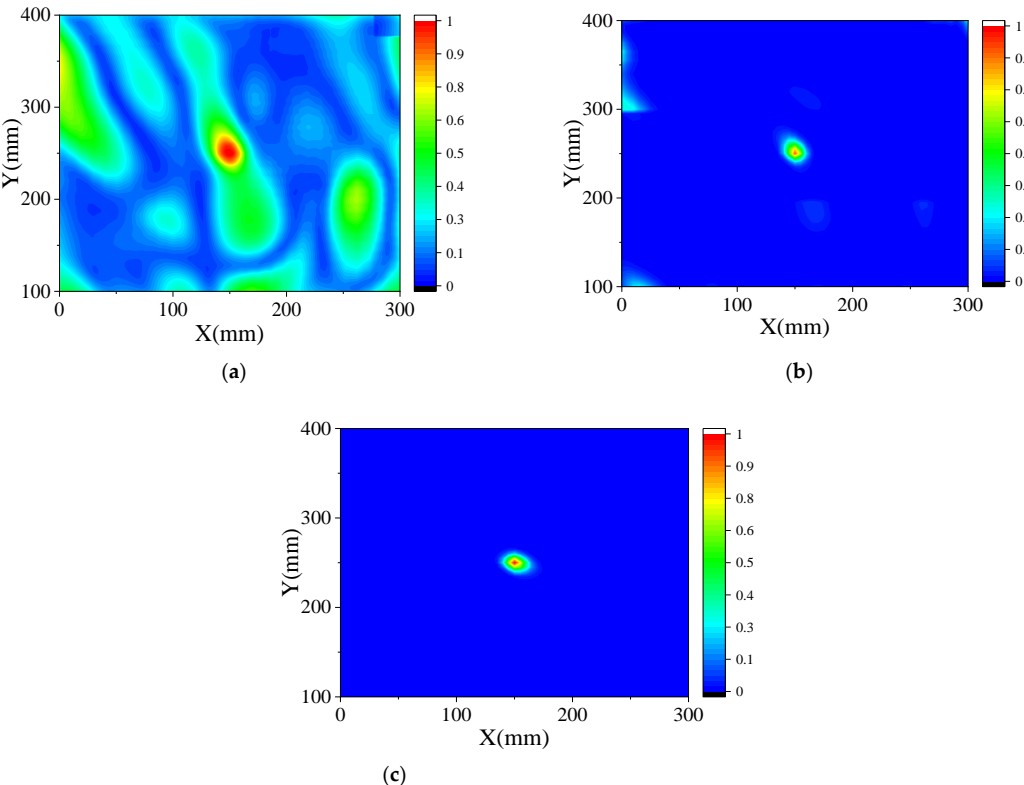

**Figure 5.** The imaging results of UWB-CS and SI-CS for a single PIM source: (**a**) UWB-CS and (**b**) SI-CS with $\delta = 0.8$ and $\mu = 0.5$ and (**c**) SI-CS with $\delta = 0.8$ and $\mu = 0.8$.

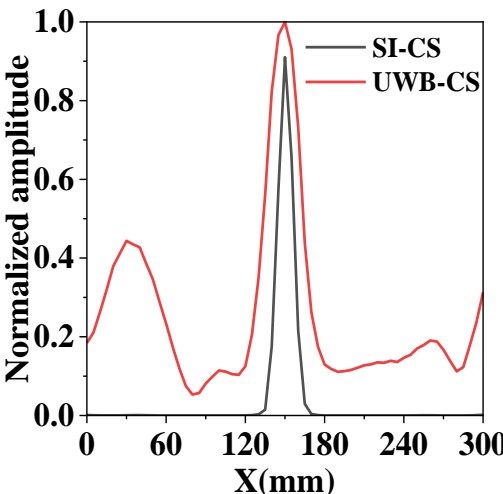

**Figure 6.** The trend of the peak in the lateral direction at the PIM source location for the UWB-CS method and the SI-CS method.

Considering the impact on positioning accuracy at different SNRs, the RMSEs corresponding to different SNRs of UWB-CS and SI-CS after multiple simulations are shown in Figure 7a. Due to the fuzzy synchronous principle, SI-CS exhibits better stability compared to UWB-CS at different SNRs.

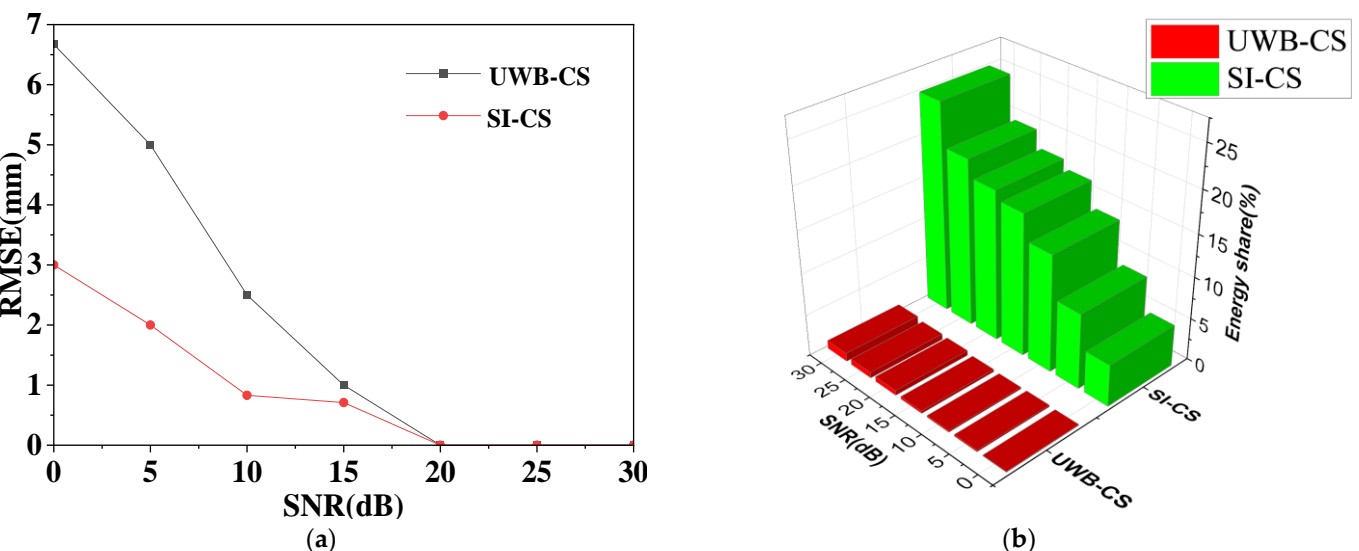

**Figure 7.** (**a**) The RMSEs of the UWB-CS method and the SI-CS method. (**b**) The energy share of the UWB-CS method and the SI-CS method.

Furthermore, considering the energy share of the PIM position in the imaging results, after several simulations and calculations, we can observe that the energy share of SI-CS is more concentrated compared to that of UWB-CS according to Equation (15) in Figure 7b. This reflects the better focusing effect of the SI-CS method. Due to the larger energy share, the better stability of the SI-CS method compared to UWB-CS is also reflected. This is helpful for the determination of the PIM source's target location.

From Figure 5c, we can know that at this time, due to the better imaging conditions, there is little effect on the imaging after the value of $\mu$ is changed.

### 4.2. Double Targets

In this example, an imaging area size of $300 \times 300$ mm$^2$ is set. The coordinates of the PIM sources are (50 mm, 200 mm) and (250 mm, 350 mm). In addition, white Gaussian noise was added to check the performance of the proposed method. The mean value of the added noise is 0 and the variance is the average power of the noise.

Based on Equation (12), $\rho_q$ at different frequency are shown in Figure 8. From the result, we can know that the difference between the transmission matrix closer to the center frequency in frequency and the transmission matrix corresponding to the center frequency is smaller, and at this time, the calculation result of parameter 1 is only around 0.1–0.3. Hence, we choose 0.6 to be $\delta$.

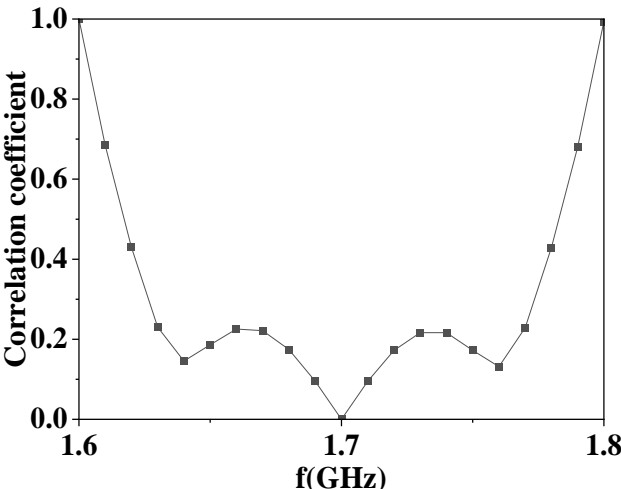

**Figure 8.** Correlation coefficient of the transmission matrix for double PIM sources.

The imaging result of the center frequency is shown in Figure 9.

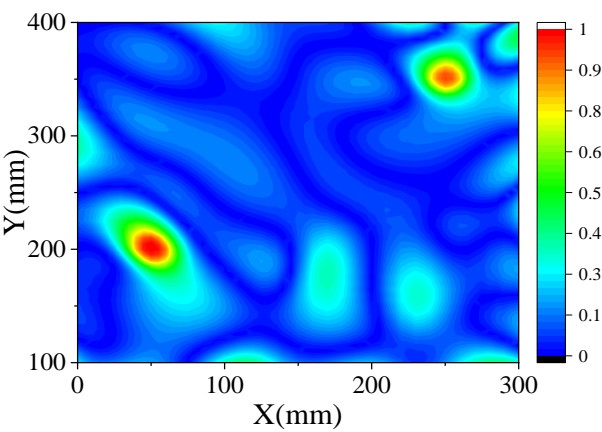

**Figure 9.** Imaging results of center frequency for double PIM sources.

As we can see in Figure 9, the imaging results are relatively flat. There are two large peak points in the imaging result and no paraflap comparable to them exists. When $\mu$ is set to 0.5, the larger paraflaps can still be screened.

The imaging results of UWB-CS and SI-CS obtained from the sensor data with SNR = 20 dB are shown in Figure 10. As we can see from the results in Figure 10a,b, both methods can form peaks at the PIM sources. However, the imaging results of the SI-CS method have a narrower range of half-peak widths at the target locations of the two PIM sources and flatter imaging results at the non-target source locations. Computation time of UWB-CS and SI-CS is 716.85 s and 174.37 s. The trend of the peak in the lateral direction

at the PIM source location for the UWB-CS method and the SI-CS method is plotted in Figure 11. By comparison, it can be found that the area of the SI-CS peak imaging section is only 25% compared to the UWB-CS method. This allows SI-CS to characterize the PIM source location more precisely.

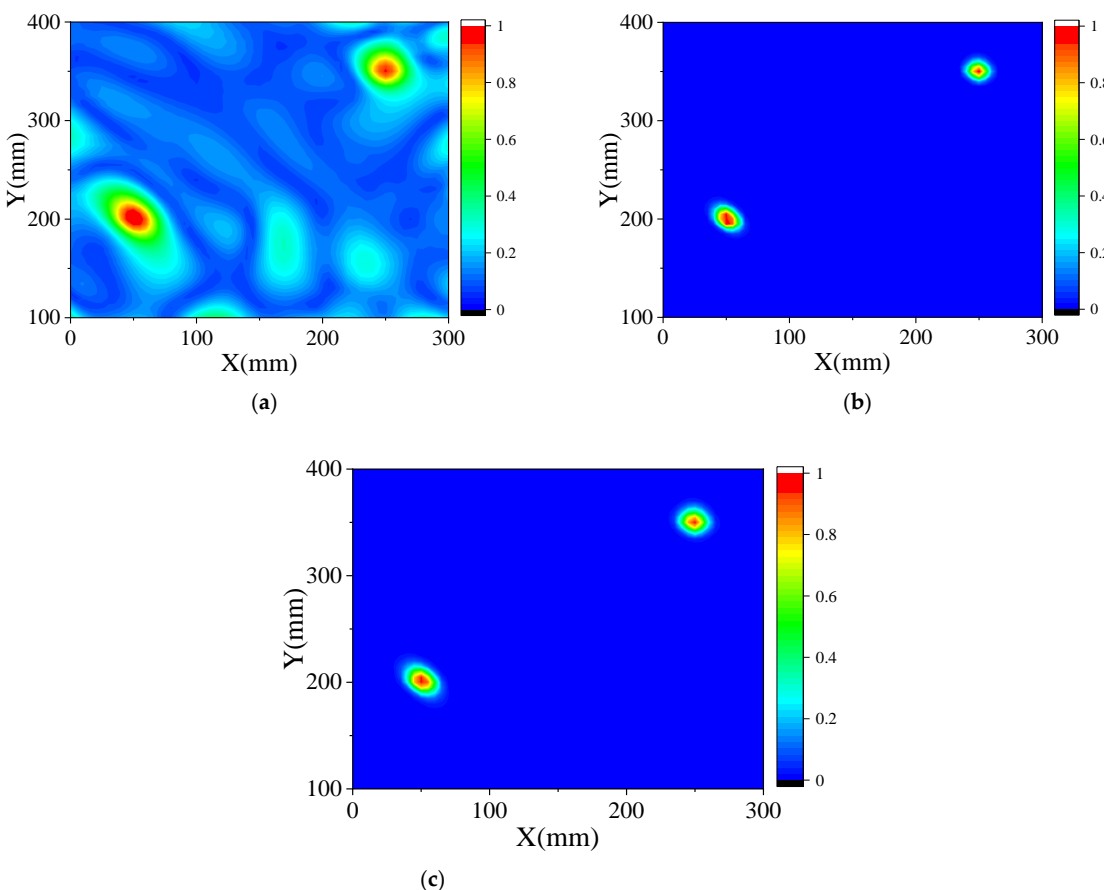

**Figure 10.** The imaging results of UWB-CS and SI-CS for double PIM sources: (**a**) UWB-CS and (**b**) SI-CS with $\delta = 0.6$ and $\mu = 0.5$ and (**c**) SI-CS with $\delta = 0.6$ and $\mu = 0.8$.

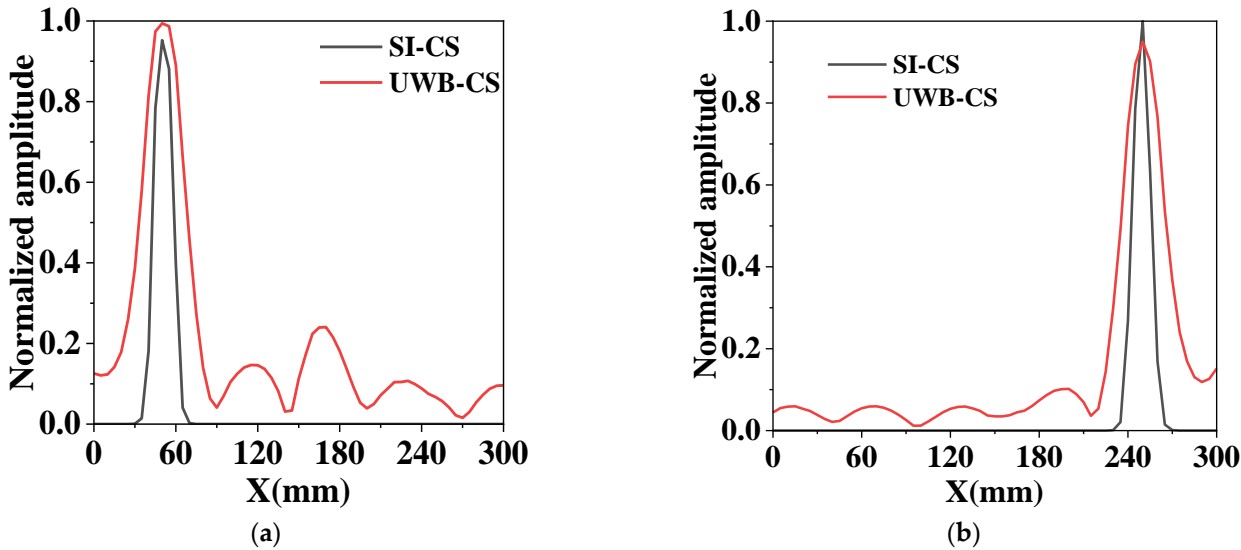

**Figure 11.** Lateral peak trends of the UWB-CS method and SI-CS method at two PIM source locations, respectively (**a**) the lateral peak trend of the first PIM source for SI-CS and UWB-CS (**b**) the lateral peak trend of the second PIM source for SI-CS and UWB-CS.

In this scenario, the results were calculated for several simulations with different SNRs, and the RMSEs of the SI-CS method as well as the UWB-CS method are shown in Figure 12a. Due to the fuzzy synchronous principle, SI-CS exhibits better stability compared to UWB-CS at different SNRs. In the case of low SNR, the SI-CS method has a lower RMSE due to the fuzzy synchronization principle. Additionally, at the same time, the RMSE of the SI-CS method converges more rapidly, and the performance is better than that of the UWB-CS method for different SNRs.

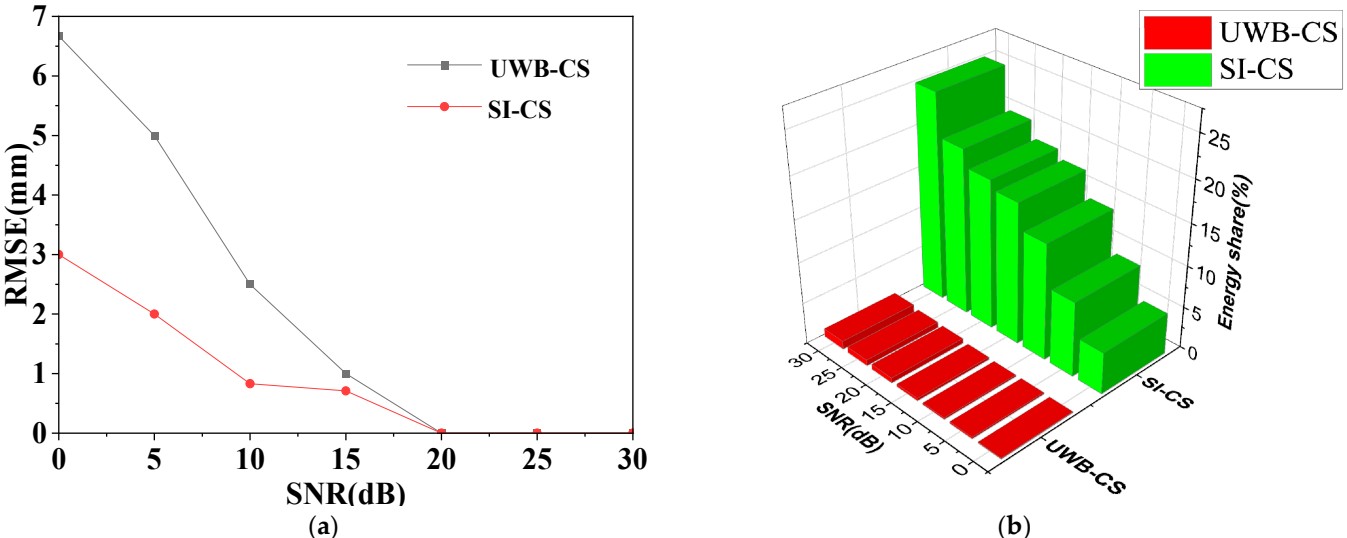

**Figure 12.** (**a**) The RMSEs of the UWB-CS method and the SI-CS method. (**b**) The energy share of the UWB-CS method and the SI-CS method.

From the data of the energy share in Figure 12b, the data of the SI-CS method at different SNRs are significantly higher than those of the UWB-CS method. The energy of the SI-CS method is mainly concentrated at the PIM sources, which demonstrates a better focusing effect. From the higher energy share of SI-CS at different SNRs, it is clear that the SI-CS method further improves the accuracy of identifying the PIM source when the SNR is low.

From Figure 10c, we can know that at this time, due to the better imaging conditions, there is little effect on the imaging after the value of $\mu$ is changed.

### 4.3. Multiple Targets

In this example, an imaging area size of $300 \times 300$ mm$^2$ is set. Additionally, four PIM sources are set in the area. The coordinates of the PIM sources are (50 mm, 200 mm), (120 mm, 270 mm), (200 mm, 200 mm), and (250 mm, 350 mm). In addition, white Gaussian noise was added to check the performance of the proposed method. The mean value of the added noise is 0 and the variance is the average power of the noise.

Based on Equation (12), $\rho_q$ values at different frequencies are shown in Figure 13. From the result, we can know that the difference between the transmission matrix closer to the center frequency in frequency and the transmission matrix corresponding to the center frequency is smaller, and at this time the calculation result of $\mu$ is only around 0.1–0.3. Hence, we choose as 0.6 to be $\delta$.

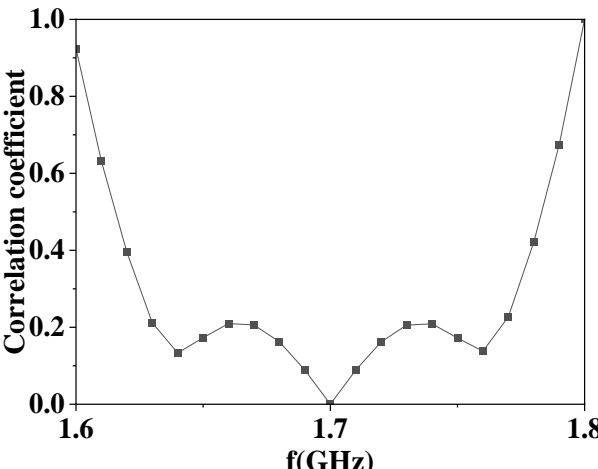

**Figure 13.** Correlation coefficient of the transmission matrix for multiple PIM sources.

The imaging result of the center frequency is shown in Figure 14.

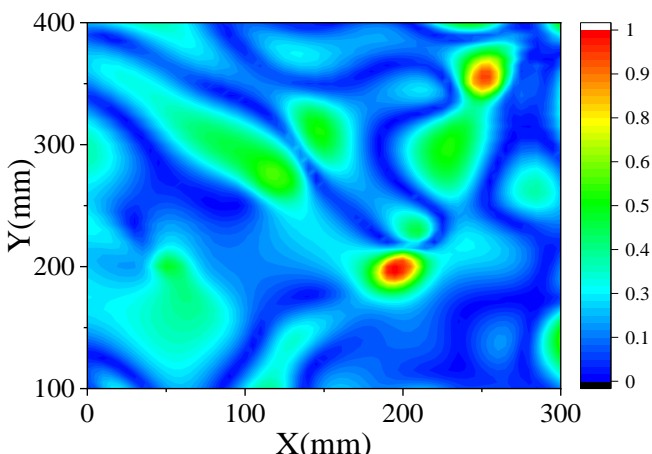

**Figure 14.** Imaging results of center frequency for multiple PIM sources.

As we can see in Figure 14, the imaging results are not flat. In addition to the two major peak points, there are many larger paraflaps present. At this point, it is not possible to determine whether there is a submerged PIM source in these larger paraflaps. In order to filter out the PIM sources, we set $\mu$ to 0.3. This value can cover all the larger paraflap positions, so as to filter out the possible PIM sources among them.

The imaging results of UWB-CS and SI-CS obtained from the sensor data with SNR = 20 dB are shown in Figure 15. From the results Figure 15a,b, we can see that under the UWB-CS method, individual PIM sources are overwhelmed at this time. However, there is still a certain peak at each PIM source in the imaging results from the SI-CS method. Due to the fuzzy synchronous principle of fuzzy synchronization, we can filter out the locations where random partials are generated, so that a peak can be successfully formed at the location of the PIM source that is swamped by the partials. The computation time of UWB-CS and SI-CS is 713.66 s and 174.40 s. The trend of each peak in the lateral direction at the PIM sources' location for the UWB-CS method and the SI-CS method is plotted in Figure 16. Again, the feature of better focusing is observed in the SI-CS method.

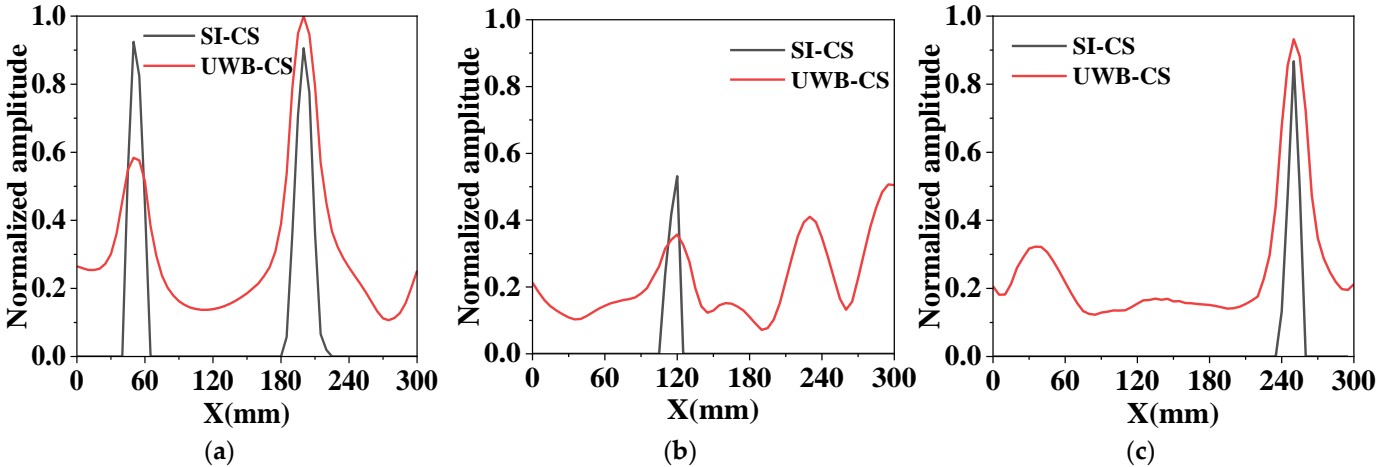

**Figure 15.** The imaging results of UWB-CS and SI-CS: (**a**) UWB-CS and (**b**) SI-CS with $\delta = 0.6$ and $\mu = 0.3$, (**c**) SI-CS with $\delta = 0.6$ and $\mu = 0.5$ and (**d**) SI-CS with $\delta = 0.6$ and $\mu = 0.8$.

**Figure 16.** The trend of the peak in the lateral direction at the PIM source location for the UWB-CS method and the SI-CS method; (**a**) the first and second PIM sources; (**b**) the third PIM source; (**c**) the fourth PIM source.

As shown in Figure 15c,d, when $\mu$ fails to cover the major paraflap, it will be difficult for the possible PIM sources that are flooded by the paraflap to be screened out in subsequent calculations.

Further, we changed the values of $M = 4$ and $N = 5$ to use a larger array to locate these four PIM sources. The results are shown in Figure 17, and we can see from the results that it was possible to accurately locate the four PIM sources after increasing the number of array cells. Comparing this result, this shows that our proposed method can also locate accurately when there are multiple PIM sources when the number of antennas is small.

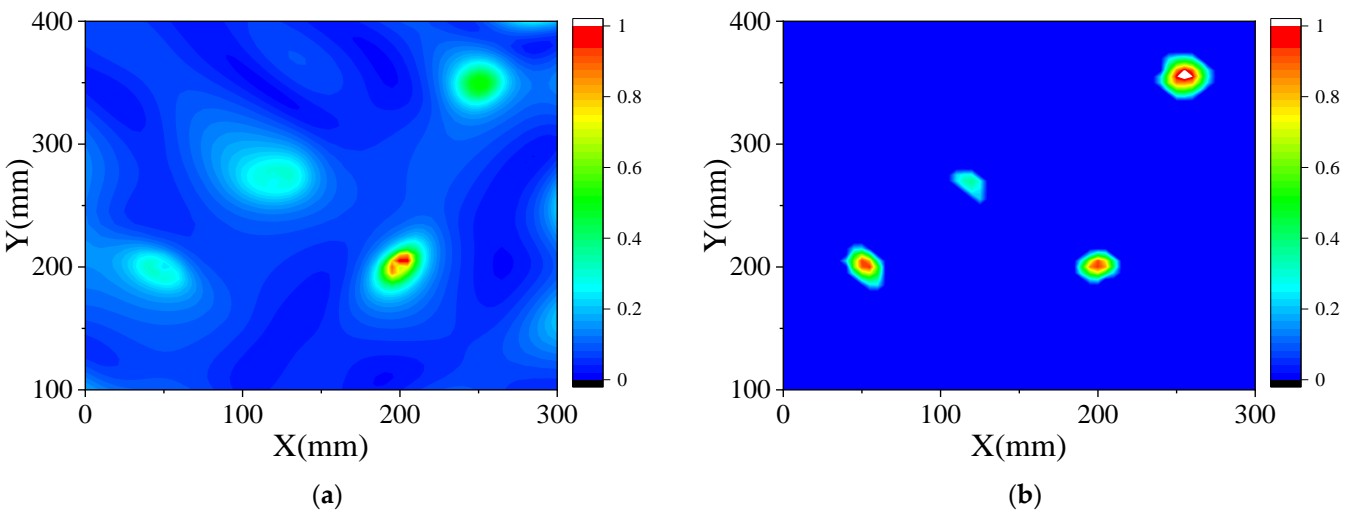

(a)

(b)

**Figure 17.** The imaging results of UWB-CS and SI-CS: (**a**) UWB-CS and (**b**) SI-CS with $\delta = 0.6$ and $\mu = 0.3$.

## 5. Conclusions

In this paper, we propose an SI-CS electromagnetic imaging method based on the expansion of the coprime array. Compared with the current UWB-CS method, the proposed method has a good imaging effect for single or multiple PIM sources and can have a 16% and 25% half-peak area for one PIM source and two PIM sources under the simulation condition. In the case of four PIM sources, the proposed method can image the already submerged PIM sources, thanks to the fuzzy simultaneity principle. At the same time, the proposed method can effectively reduce the size of the secondary flap in the imaging, and has a lower RMSE and a higher energy share than UWB-CS under different SNR conditions, which means it can determine the location of the PIM source target more effectively.

**Author Contributions:** Methodology, M.L., F.L. and D.Z.; Validation, M.M.; Writing—original draft, S.L., Z.C. and X.L. All authors have read and agreed to the published version of the manuscript.

**Funding:** This research was funded by the National Natural Science Foundation of China (NSFC) grant number no. 61901087, China Postdoctoral Science Foundation grant number no. 2019M663471, "2021 Medical Oncology and Engineering Innovation Fund" project grant number no. ZYGX2021YGCX008.

**Data Availability Statement:** Not applicable.

**Conflicts of Interest:** The authors declare no conflict of interest.

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
