# Peer review of "Electromagnetic Imaging of Passive Intermodulation Sources Based on Virtual Array Expansion Synchronous Imaging Compressed Sensing"

_electronics, doi:10.3390/electronics12071653_

Round 1

Reviewer 1 Report

Liu et al have demonstrated a synchronous imaging compressed sensing (SI-CS) method for passive intermodulation (PIM) sources imaging. By filtering out unwanted random noises in the frequency domain below a set threshold, and renormalizing signals separately in each screened range, the authors are able to achieve higher localization accuracy, avoid flooding problem, and reduce computation efforts in PIM sources detection. In general, this manuscript is nicely written with claims supported by analytical and numerical data. I however need the authors to address my following concerns before I can recommend acceptance for publication in the MDPI Electronics journal.

The key step of the current work is to choose proper threshold values for correlation coefficient δ and covariate for signal power μ. In the 3 results shown by the authors, covering the situations with single-source PIM, double-source PIM, and triple-source PIM, I noticed that the authors have chosen the following 3 sets of threshold values: (δ, μ) = (0.8, 0.5) for 1 PIM source, (0.6, 0.5) for 2 PIM sources, and (0.6, 0.3) for 3 PIM sources. Can the authors elaborate on how these different values are chosen? I recommend the authors to include imaging results for the same PIM configuration, but with different threshold values in the same figure. This will help the readers understand how the threshold values affect the imaging quality.

The authors mentioned that their proposed method reduces computation resources. I recommend the authors to show direct evidence to support this claim. For example, to achieve the same PIM source detection accuracy, how much more computation time would other method require compared to the current SI-CS method?

Reviewer 2 Report

The paper proposes to use compressed sensing to localize radial passive intermodulations. The idea is to use compressed sensing to overcome the challenge of having only a limited number of antennas for measurement because compressed sensing enables reconstructing a source signal using a limited number of samples. Experiments are offered to demonstrate that the proposed method is effective.     This is an application of CS in a new area and is interesting. I have the following comments to be addressed for the next round of reviews.       1. Compressed sensing has been used in other applications to reduce the number of measurement devices:   a. Nikitaki, S. and Tsakalides, P., 2011, August. Localization in wireless networks based on jointly compressed sensing. In 2011 19th European Signal Processing Conference (pp. 1809-1813). IEEE.   b. Guibène, W. and Slock, D., 2013. Cooperative spectrum sensing and localization in cognitive radio systems using compressed sensing. Journal of Sensors2013.   c. Rostami, M., Cheung, N.M. and Quek, T.Q., 2013, May. Compressed sensing of diffusion fields under heat equation constraint. In 2013 IEEE International Conference on Acoustics, Speech and Signal Processing (pp. 4271-4274). IEEE.   d. Ranieri, J., Chebira, A., Lu, Y.M. and Vetterli, M., 2011, May. Sampling and reconstructing diffusion fields with localized sources. In 2011 IEEE International Conference on Acoustics, Speech and Signal Processing (ICASSP) (pp. 4016-4019). IEEE.   The above work should be discussed in the introduction section to provide a complete context about this work.   2. Could you add more justifications about the choices that you use for hyperparameters in the experiments? It would be better to provide a perspective about this choices.     3. Could you also perform experiments and vary the value for M and N to study how the performance is affected?   4. Could you add a comparison of the solution obtained by Eq 17 with more classic methods to make the advantage of the proposed method more clear?     5. In the experiments because the noise is added synthetically, experiments should be performed several times and both the mean and the standard deviation should be reported.   6. Please arrange to release the code on a public domain such as GitHub to allow other researchers to reproduce the results easily.

Round 2

Reviewer 2 Report

The authors have addressed my concerns decently.